# Subject-Dependent Artifact Removal for Enhancing Motor Imagery Classifier Performance under Poor Skills

**DOI:** 10.3390/s22155771

**Published:** 2022-08-02

**Authors:** Mateo Tobón-Henao, Andrés Álvarez-Meza, Germán Castellanos-Domínguez

**Affiliations:** Signal Processing and Recognition Group, Universidad Nacional de Colombia, Manizales 170003, Colombia; amalvarezme@unal.edu.co (A.Á.-M.); cgcastellanosd@unal.edu.co (G.C.-D.)

**Keywords:** Brain-Computer Interface, electroencephalography, motor imagery, artifact removal, functional connectivity

## Abstract

The Electroencephalography (EEG)-based motor imagery (MI) paradigm is one of the most studied technologies for Brain-Computer Interface (BCI) development. Still, the low Signal-to-Noise Ratio (SNR) poses a challenge when constructing EEG-based BCI systems. Moreover, the non-stationary and nonlinear signal issues, the low-spatial data resolution, and the inter- and intra-subject variability hamper the extraction of discriminant features. Indeed, subjects with poor motor skills have difficulties in practicing MI tasks against low SNR scenarios. Here, we propose a subject-dependent preprocessing approach that includes the well-known Surface Laplacian Filtering and Independent Component Analysis algorithms to remove signal artifacts based on the MI performance. In addition, power- and phase-based functional connectivity measures are studied to extract relevant and interpretable patterns and identify subjects of inefficency. As a result, our proposal, Subject-dependent Artifact Removal (SD-AR), improves the MI classification performance in subjects with poor motor skills. Consequently, electrooculography and volume-conduction EEG artifacts are mitigated within a functional connectivity feature-extraction strategy, which favors the classification performance of a straightforward linear classifier.

## 1. Introduction

The cognitive ability of motor imagery (MI) is one of the most active areas of Brain-Computer Interface (BCI) research; it allows handling external devices merely by imagining movement, without the involvement of peripheral nerves [1], having applications in motor function rehabilitation [2,3], and motor function assistance [4], among others [5]. Furthermore, other Human Machine Interfaces (HMI) have been proposed to boost BCI systems. For example, alongside BCI, eye-tracking-based approaches have provided a strong coupling between cognitive psychological attention tests and attention levels determined by a BCI [6]. In addition, several studies compare brain activity-based attention with the continuous performance and variables of attention tests, a.k.a. CPT and TOVA [7]. On the other hand, some interesting results are also obtained when applying deep learning techniques in inversion problems for eye-tracking [8,9].

Concerning MI-based paradigms, Electroencephalography (EEG) is widely used to record brain neural activity because of its high time resolution, portability, and cost-effectiveness compared to other neuroimaging methods [10]. However, capturing neural activity from the scalp faces several restrictions: non-stationarity and nonlinearity of EEG data [11], a low-spatial resolution that may also affect the time-resolution [12], and the inter- and intra-subject variability for which the distribution of features extracted from time-variant brain patterns across subjects can be different for the same tasks [13]. All those factors result in low Signal-to-Noise Ratio (SNR) phenomena, posing a challenge in EEG analysis [14,15,16]. SNR values, in general, are affected by several factors [17]. At first, the volume conduction effect leads to low spatial resolution when the same electric fields generated at a single brain location propagate through tissues and are detected by more than one sensor [18]. Hence, EEG channels are spatially associated with one another, resulting in incorrect source information on the cortex, and topographical components spread spatially (the field spread effect). Another factor leading to low SNR values is that the sensor montage captures unwanted signals, which mix with the brain activity information [19]. To be more precise, involuntary muscle contraction sources or artifacts are formed in the human body by diverse biosignals such as electrooculography (EOG), electrocardiography (ECG), scalp electromyography (EMG), heart-related pulsatile motions, and respiration [20]. An additional factor associated with low SNR is the significant difference in the subject’s capacity for sustained engagement in attention-demanding tasks (termed subject inefficiency phenomenon [21]). Typically, about 15–30% of users cannot master MI systems, eliciting brain activation patterns with very high inter-session variability of feature distributions that are difficult to classify successfully (with accuracy lower than 70% [22,23]).

In practice, simple classification algorithms that follow signal processing techniques to increase the SNR ratio are employed to classify MI tasks since they are fast training, have lower computational complexity, provide a better explainability, and demand light data techniques, such as spatial filtering [24]. Thus, a widely-used approach to reducing adverse effects consists of subtracting artifacts from the contaminated EEG data using artifact templates or uncorrelated reference signals [25,26]. To this end, spatial filters based on blind source-separation methods are employed mainly through Independent Component Analysis (ICA), which rejects those independent sources with a proximate structure to the artifacts [27]. Another approach to artifact subtracting is the removal of average electrical activity due to the volume conduction effect measured across neighboring sensors, using spatial filtering techniques such as Common Average Reference [28] or Surface Laplacian (SL) [29]. Approaches combining ICA with SL have also been recently suggested for artifact removal [30]. However, their evaluation to improve the SNR in individuals with poor skills in practicing MI tasks remains challenging [31]. In particular, subject inefficiency and variability in feature distributions indicate that assuming larger amplitudes or independent spectra to separate brain activity signals and artifacts is unlikely to be effective [32].

In this paper, we propose an approach for subject-dependent preprocessing to reduce the low SNR phenomenon in subjects performing poorly in motor imagery tasks by selectively removing the artifacts according to the classifier accuracy achieved by each individual. The preprocessing methods are Surface Laplacian Filtering and Independent Component Analysis to remove artifacts related to EOG signals and the volume conduction effect. Nonetheless, instead of relying on commonly-used spatial patterns that extract mostly event-related desynchronization features [33], several functional connectivity measures of spatially-distributed regions are investigated to extract motor imagery features and identify inefficiency subjects. The proposed approach of Subject-dependent Artifact Removal, termed *SD-AR*, has been evaluated on two real-world databases, demonstrating its ability to tackle the influence of artifacts and intertrial/inter-subject amplitude variability, improving the classifier performance and enhancing interpretability in subjects with poor motor imagery skills.

The rest of the paper is organized as follows: Section 2 briefly discusses the spatial filtering methods used for artifact removal. Section 3 defines the experimental set-up, including a description of both MI datasets evaluated and the parameter tuning of preprocessing filtering and feature extraction methods. Section 4 presents the results of classifier performance achieved by the whole subject set and the group of poor-performing individuals, including the interpretability of brain neural responses elicited by the MI paradigm. Lastly, Section 5 discusses the conclusions from the presented results, including future projections.

## 2. Methods

The main mathematical fundamentals of the well-known Surface Laplacian filtering method are given. Then, the Independent Component Analysis technique is described. Finally, we outline our Subject-dependent Artifact Removal *SD-AR* approach. SD-AR allows building a discriminant preprocessing strategy based on Surface Laplacian filtering and Independent Component Analysis to feed a Functional Connectivity-based feature extraction stage for further classification.

### 2.1. Surface Laplacian Filtering

Let $X\in R^{C\times T}$ be an EEG signal with C∈N,T∈N being the number of channels and time samples, respectively. The Surface Laplacian computes the second spatial derivative of the underlying flow of electrical current produced by brain activity at the electrode c∈C concerning the neighboring scalp potentials $c^{\prime }\in C,\;c\neq c^{\prime }$. To this end, spherical splines are employed to project the sensors’ scalp positions onto a sphere, interpolating their actual electric potentials iteratively on the new coordinates via Legendre Polynomials with the elements: (1)$$\begin{array}{cc} p(c,c^{\prime }) & =\frac{1}{4\pi }\sum_{n\in o}\frac{\alpha \left(2n+1\right)P_{n}\left(\mathrm{cosdist}\;\left(e_{c},e_{c^{\prime }}\right)\right)}{{\left(n\left(n+1\right)\right)}^{\rho -\alpha }} \end{array}$$
where $P_{n}$ is the Legendre Polynomial of order *n*, o∈N is the highest polynomial order considered, $\rho \in R^{+}$ is a smoothness constant, $\mathrm{cosdist}\;\left(e_{c},e_{c^{\prime }}\right)=1-{∥e_{c}-e_{c^{\prime }}∥}_{2}^{2}/2$ is the cosine distance between a pair of electrode positions, and $e_{c},e_{c^{\prime }}\in R^{3}\left[-1,1\right]$ are the electrode positions normalized to a unit-radius sphere. The notation ${∥\cdot ∥}_{2}$ stands for $\ell_{2}$-norm distance.

Thus, the Laplacian filter of EGG data, $X_{L}\in R^{C\times T}$, is computed as [34]: $$\begin{array}{cc} X_{L} & =H{\left(X^{⊤}G_{s}^{-1}-X^{⊤}G_{s}^{-1}1G_{s}^{-1}/1G_{s}^{-1}1\right)}^{⊤} \end{array}$$
where $1\in N^{C\times C}$ is an all-ones matrix, and $G_{s}=G+\lambda I$ is a smoothed version of G, $I\in R^{C\times C}$ is the identity matrix, λ∈R[0,1] is a regularization parameter; and G and $H\in R^{C\times C}$ are the weighting matrices that hold elements expressed in terms of parameter α in Equation (Equation 1), respectively, as below: $$\begin{array}{cc} \alpha & =\left\{\begin{array}{cc} \;1, & p\left(c,c^{\prime }\right)=g\left(c,c^{\prime }\right)\\ -1, & p\left(c,c^{\prime }\right)=h\left(c,c^{\prime }\right). \end{array}\right. \end{array}$$

### 2.2. Independent Component Analysis

Brain and artifact sources are assumed to mix linearly in X=WS, where $W\in R^{C\times Q}$ is the mixing matrix and $S\in R^{Q\times T}$ is the sources’ matrix, being Q≤C the number of sources. Under the assumption that the components are statistically independent and non-Gaussian distributed, the joint estimation of $\hat{S}=\left\{{\hat{s}}_{q}:q\in Q\right\}$ and $\hat{W}=\left\{{\hat{w}}_{q}:q\in Q\right\}$ can be performed through the negentropy-based optimizing framework as follows [35]: (2)$$\begin{array}{cc} {\hat{w}}_{q}=\underset{w_{q}}{\mathrm{argmax}} & \;H\left(s_{\nu }\right)-H\left(w_{q}^{⊤}X\right),\;\forall q\in Q\\ \; & \;s.t.:\;{∥w_{q}∥}_{2}=1 \end{array}$$
where ${\hat{w}}_{q}\in R^{C}$ is a column vector of unmixing weights computed for a row source, $s_{\nu }\;∼\;N\left(0,1\right)$ is a zero-mean, unit-variance Gaussian random variable ($s_{\nu }\in R^{T}$) and $H(\nu )=E\left\{-\log(p(\nu ))\;:\;\forall \nu \right\}$ is the Shannon entropy. The notation $E\left\{\;:\;\forall \nu \right\}$ stands for expectation operator across a random variable ν.

Consequently, the maximization procedure described above is performed over every q∈Q component. Here, *Q* is fixed to *C*, i.e., the number of EEG-montage channels. Therefore, to recover the sources and separate them between brain or artifact signals [36], the unmixing model yields: (3)$$\begin{array}{cc} \hat{S} & ={\hat{W}}^{⊤}X. \end{array}$$

### 2.3. Subject-Dependent Artifact Removal (SD-AR)

In the presence of a reference electrooculography signal $a_{q^{\prime }}\in R^{T},\;q^{\prime }\in Q^{{}^{\prime }}$, we use a pairwise similarity estimate to remove artifacts from the contaminated EEG data by comparing an ICA source with every available reference. Thus, each ${\hat{s}}_{q}$ component is rejected or selected according to the following thresholding rule: (4)$$\begin{array}{cc} {\tilde{s}}_{q}\; & =\left\{\begin{array}{cc} {\hat{s}}_{q}, & |E\left\{\left({\hat{s}}_{q}-\mu_{{\hat{s}}_{q}}\right)\left(a_{q{}^{\prime }}-\mu_{a_{q{}^{\prime }}}\right)\right\}|/\sigma_{{\hat{s}}_{q}}\sigma_{a_{q^{\prime }}}\leq \gamma \\ 0, & \mathrm{Otherwise} \end{array}\right. \end{array}$$
where $\mu_{{\hat{s}}_{q}},\mu_{a_{q{}^{\prime }}}$, $\sigma_{{\hat{s}}_{q}},\sigma_{a_{q{}^{\prime }}}\in R$ are the values of the mean and standard deviation computed for ${\hat{s}}_{q}$ and $a_{q^{{}^{\prime }}}$, respectively; and $\gamma \in R^{+}$ is the threshold established to remove (above) or select (under) the ICA components. The notation |·| stands for absolute value. Hence, the cleaned signal is reconstructed from the unmixing model in Equation (Equation 3). That is, $\tilde{X}\in \hat{W}\tilde{S}$, where $\tilde{S}\in R^{Q\times T}$ is the selected ICA matrix that removes the most closely ICA sources related to the reference artifacts.

In addition, applying both Surface Laplacian Filtering and Independent Component Analysis approaches to EOG-artifact removal and volume conduction effect relies on the principle of a similar influence over the whole set of individuals. Instead, for subject-dependent effectiveness, we select either method of preprocessing based on the quality attained by each individual. For this purpose, the *SD-AR* approach introduces the selector coefficient, noted as $\eta_{m}\in N\left[0,1,2,3\right]$, that gets the value according to the following scenarios of maximal accuracy performed by each m∈M subject: (5)$$\begin{array}{cc} \eta_{m} & =\left\{\begin{array}{c} 0,\;Y=X\\ 1,\;Y=\tilde{X}\\ 2,\;Y=X_{L}\\ 3,\;Y={\tilde{X}}_{L} \end{array}\right. \end{array}$$
where ${\tilde{X}}_{L}$ denotes the sequential application of both preprocessing procedures.

The classifier performance is computed using Functional Connectivity (FC) as the feature extraction method. In particular, the following FC measurements extracted on a trial basis are considered:-*Power-based connectivity* correlates between couples of electrodes over time founded on power fluctuations. Regarding this FC, the following pairwise measures are evaluated [37,38,39]:
(6a)$$\begin{array}{cc} \mathrm{Pearson}\mathrm{Correlation}= & E\left\{\left(x-\mu_{x}\right)\left(x^{\prime }-\mu_{x^{\prime }}\right)\right\}/\sigma_{x}\sigma_{x^{\prime }} \end{array}$$
(6b)$$\begin{array}{cc} \mathrm{Motifs}= & \;E\left\{\times i\left({\bar{x}}_{b}={\bar{x}}_{b}^{\prime }\right):\forall b\in B\right\} \end{array}$$
(6c)$$\begin{array}{cc} \mathrm{Gaussian}\mathrm{FC}= & E\left\{\exp(-{∥x-x^{\prime }∥}_{2}^{2}/2\sigma^{2})\right\} \end{array}$$
where $x,x^{\prime }\in R^{T}$ are the signals captured at a pair of electrodes, $\mu_{x},\mu_{x^{\prime }}$, $\sigma_{x},\sigma_{x^{\prime }}\in R$ are the values of mean and standard deviation estimated for $x,x^{\prime }$, respectively; $\bar{x},{\bar{x}}^{\prime }$ are the motifs series calculated using the synchronization method (each one lasting B≤T), $\times i\left({\bar{x}}_{b}={\bar{x}}_{b}^{\prime }\right)$ is the vector of coincidences between the motif series’ elements; and $\sigma \in R^{+}$ is the kernel bandwidth needed for Gaussian FC.-Functional Connectivity based on *Phase Coupling* between two electrodes. The following measures are evaluated [40,41]:
(7a)$$\begin{array}{cc} \mathrm{Spectral}\mathrm{Coherence}= & \frac{|E\left\{\Xi_{xx^{\prime }}\right\}|}{\sqrt{E\left\{\Xi_{xx}\right\}E\left\{\Xi_{x^{\prime }x^{\prime }}\right\}}} \end{array}$$
(7b)$$\begin{array}{cc} \mathrm{Phase}-\mathrm{Locking}\mathrm{Value}= & |E\left\{\Xi_{xx^{\prime }}/|\Xi_{xx^{\prime }}|\right\}| \end{array}$$
where $\Xi_{xx^{\prime }}\in R^{T}$ is the cross-spectral density between x and $x^{\prime }$; while $\Xi_{xx}$, $\Xi_{x^{\prime }x^{\prime }}\in R^{T}$ are the estimates of power spectral density computed for x and $x^{\prime }$, respectively.

In the above-considered FC measurements, the expectation operator averages across the whole time set of each trial.

## 3. Experimental Set-Up

The proposed methodology of selective preprocessing is based on the well-known methods of Surface Laplacian filtering and Independent Component Analysis. Namely, our SD-AR automatically switches each one of the techniques mentioned above concerning the BCI accuracy score. In this sense, SD-AR aims to improve each subject’s classification performance and the interpretability of elicited brain activity responses along groups of subjects with different skills and abilities for MI practice. As shown in Figure 1, the tested preprocessing approach appraises the following stages:(i)Subject-dependent preprocessing for artifact removal (*SD-AR*). According to the highest individual classifier accuracy, removing artifacts over each subject is conducted according to one of the following combinations of the ICA and Laplacian filters in Equation (Equation 5): both filters, one only, or neither. In applying both filters sequentially, the ICA procedure is first performed before the Laplacian algorithm, as suggested in [42]. For comparison, two additional cases of preprocessing are also evaluated: subject-independent artifact removal (*Ind-AR*) using both filters of artifact subtraction over every individual regardless of his achieved classifier performance, and no removal filtering (*Raw*) of all acquired EEG data.(ii)Feature Extraction for testing the power-based and phase-based FC measurements. As suggested in [43], the set of EEG signals is band-pass filtered within the following four frequency bandwidths (rhythms): {μ∈[8–12],
$\beta_{l}\in \left[12–15\right],$
$\beta_{m}\in \left[15–20\right]$, and $\beta_{h}\in \left[18–40\right]\}$ Hz. The features are extracted within the time window of post cue onset. That is, 0.5–3.5 s for DBI and 0.5–2.5 s for DBII (see Section 3.1). The feature extraction is built for each label by the vectorized version of the upper triangular matrix, sizing C×(C−1)/2, computed over each trial set for each FC measure under evaluation. The obtained super vectors within the four considered rhythms are further concatenated to create a single vector, of size 4C×(C−1)/2, to be fed into the classifier. Regarding the FC parameters, the following values are specified: The kernel bandwidth in GFC, σ, is fixed as the median averaged over the input distances [44]; and the degree of motifs is set to 3 while the lag to 1, as suggested in [45].(iii)Estimating classification performance using a linear discriminant analysis (LDA) algorithm evaluated through a 10-fold cross-validation strategy over the training set provided for DBI and a simple ten-iteration 20–80% training-testing split for DBII. Of note, DBI provides a predefined testing set, which is used to report the final performance. Note that the preprocessing parameters are tuned at this stage according to the achieved classifier performance.

All experiments were carried out in Python 3.8, with the sklearn and MNE libraries, on a Google Collaboratory environment (code repository: https://github.com/mtobonh/SD-AR, accessed on 30 June 2022).

### 3.1. EEG Data Description

To appraise the properties of *SD-AR*, we test the following databases of motor-related tasks, having EEG data contaminated with EOG artifacts and affected by the volume conduction effect:

**DBI**: BCI Competition 2008 - Graz Dataset 2a is a public collection available at (http://www.bbci.de/competition/iv/index.html accessed on 1 April 2022) that holds EEG data from M=9 subjects. The data were acquired in two sessions on different days within a MI paradigm, consisting of four motor imagery tasks: imagining the movement of the left hand, the right hand, both feet, or the tongue. Each session comprised six runs with 48 trials (12 for each of the four possible classes), yielding R=288 trials per session. A short acoustic warning and a cross on a black screen indicated the beginning of each trial, lasting T=7 s. At the time of 2 s, a visual cue appeared on the screen for a period of 1.25 s (an arrow pointing left, right, down, or up, corresponding to one of the four MI tasks). The cue instructed the subject to perform the indicated MI task until the cross had disappeared from the screen at 6 s. In the end, a short break followed, and the screen went black.

EEG data were recorded using a 22-channel montage with Ag/AgCl electrodes, according to the 10/20 system (C=22). In addition, three EOG electrodes were used to record ocular artifacts. Both EEG and EOG signals were sampled at a sampling rate of 250 Hz and bandpass-filtered between the range of 0.5 to 100 Hz. A 50 Hz Notch filter is also applied. The subjects’ datasets are stored in the General Data Format for biomedical signals, using one file per subject and session.

**DBII**: This collection is publicly available at (http://gigadb.org/dataset/100295 accessed on 1 April 2022), and it holds EEG data from 52 healthy subjects (although only M=50 are available for evaluation). The data were acquired in one session according to the BCI experimental paradigm of MI with two classes (left and right hands). Every session included five or six runs, performing 100 or 120 trials per class. Each trial lasted T=7 s and began with a black screen with a fixation cross within 2 s. Next, a cue instruction appeared randomly on the screen within 3 s and prompted the subject to perform the indicated MI task. In the end, a blank screen reappeared, followed by a break between 4.1 to 4.8 s.

The EEG data were collected through C=64 Ag/AgCl electrodes, placed according to the 10/10 international system, using a Biosemi ActiveTwo system. The subjects’ datasets were sampled at 512 Hz and are stored in (*.mat) format. Aside from the MI recordings, real left-hand and right-hand movements were also collected, along with six types of noise, including blinking eyes, eyeball movement up/down, eyeball movement left/right, head movement, jaw clenching, and resting state.

### 3.2. Computation of ICA Decomposition

Initially, a five-order high pass Butterworth filter of 1 Hz is performed to remove low-frequency drifts affecting the quality of the ICA filtering algorithm. In addition, the unmixing matrix W is also orthogonalized via the whitening procedure to improve the accuracy of independent components, which are estimated by a fast ICA algorithm. Accordingly, the cost function in Equation (Equation 2) is approximated as follows [46]: $$\begin{array}{c} H\left(s_{\nu }\right)-H\left(w_{q}^{⊤}X\right)\propto {\left(E\left\{-\exp\left(-{\left(w_{q}^{⊤}X\right)}^{2}/2\right)\right\}-E\left\{-\exp\left(-s_{\nu }^{2}/2\right)\right\}\right)}^{2}. \end{array}$$

As an illustrative example, Figure 2 shows the channels-noise ratio computed as the median of normalized unmixing matrix weights of sources associated with artifacts. The median value is calculated over the subjects’ noisy trials. As seen in both evaluated databases, the ocular artifacts affect the frontal channels because of their proximity to the eyes. However, the reconstructed topoplots show that DBI has a larger effect on brain area since the EEG montage holds the third set of electrodes, as does DBII. Moreover, the small montage may produce significant variations in the estimated weights of the electrodes of frontal and centro-parietal areas.

The source selection is based on the similarity between every ICA source and the electrooculogram available for DBI. Then, the similarity is measured on a trial basis over the whole subject set through the Pearson Correlation value. Since DBII does not provide the EOG data, the similarity with the ICA components is assessed using the three frontal electrodes, as suggested in [47]. Across the subject set, the correlation values have a Gaussian probability density function, as shown in the right column of Figure 2 for either database. Therefore, the rule for source removal is fixed to the 3sigma-level computed over the *Z*-scored values (see red dashed lines). Note that this procedure is twice performed.

### 3.3. Impact of Surface Laplacian Filtering

We assess the influence of Surface Laplacian Filtering on the connectivity measurements. To this end, we adjust the parameters of the surface Laplacian filter as suggested in [48]: the highest Legendre polynomial order *o* to 10, the smoothing constant ρ to 4, and the regularization parameter λ to $1\times 10^{-5}$. Further, we conduct the testing methodology developed in [49] that explores whether the amount of FC estimates decreases after applying the spatial filtering over a specific electrode. Consequently, we check if the number of false links drops due to the field spread effect.

Specifically, we examine the functional connectivity between the Cz channel, placed over the sensorimotor area, and the remaining electrodes. For illustration’s sake, Figure 3 presents the FC matrix extracted using the pairwise Pearson correlation in Equation (Equation 6), although other metrics yield similar results. We show the corresponding topogram estimated for the Cz channel along with the FC matrix. Both FC representations are obtained for DBI (top row) and DBII (bottom row) by averaging across the trial and individual sets. As can be seen, the left-side heatmap matrix extracted from the raw data visualizes many more relationships between electrodes so that the resulting topoplots exhibit neural responses spread throughout the scalp surface.

Consequently, the MI rehearsing is masked by a high field spread effect. On the contrary, the Surface Laplacian filter impacts the functional connectivities calculation positively, reducing considerably the number of links outside the sensorimotor area activated by MI, as shown in the connectivity matrix. Furthermore, the estimated topoplot presents a focused neural activity neighboring the Cz electrode with a reduced amount of spurious connectivities caused by volume conduction. The EEG montage of DBII with a much higher number of electrodes can better illustrate this situation.

## 4. Results and Discussion

By using the functional connectivity estimators above-described in Section 2.3, the proposed methodology is evaluated in terms of the performance of each feature extraction using the following metrics: Accuracy (ACC), Cohen’s Kappa coefficient (kappa), and the Area Under the Receiver Operating Characteristics Curve (AUC) [50].

In all performance metrics, we show their testing set values of the mean ± standard deviation averaged across two different evaluating strategies: (i) Global assessment by averaging over the whole set of individuals; (ii) Group-level assessment by averaging over a concrete category of subjects. Namely, we are interested in evaluating the effectiveness of the suggested preprocessing methodology for enhancing the classifier performance of the so-called inefficiency individuals [51].

### 4.1. Classifier Performance Achieved by Subject Set

Here, we estimate the performance in three training scenarios for classifying MI tasks: (a) the classifier is fed by EEG data without preprocessing (left column—*Raw*), (b) EEG data after both preprocessing procedures for Independent Component Analysis and Laplacian Filtering (center column—*Ind-AR*), (c) the suggested preprocessing approach that switches either procedure selectively as explained before (right column—*SD-AR*).

Table 1 presents the results achieved by the evaluation set of DBI, for which the best performance values of each estimator are marked in bold. Overall, both power-based and phase-based connectivity measurements achieve poor results in the four-class classification task of DBI, with the GFC estimator being the best performer. As can be seen, training solely with raw data underperforms the proposal regardless of the FC metric employed. Despite this, *Ind-AR* is the best preprocessing scenario, even though the suggested method gets close results when compared. This weak impact of the proposal may be explained by the fact that the DBI collection faces several troubles during validation: First, the training and testing data distributions differ considerably. Figure 4 displays the number of trials strongly affected by artifacts, which have been identified using the ICA-based procedure in the training and testing datasets. As can be observed, there is a high disparity in four subjects (labeled as #2, # 5,# 1, and #3). DBI is a relatively small data set, so four subjects would be more than 40% of the data set. Secondly, the used EEG montage holds a relatively small number of channels, posing a significant restriction for SL procedures, as discussed in [52]. Consequently, validation is adversely affected by all of these issues, resulting in a low classifier performance.

On the other hand, Table 2 displays the resulting performance obtained through a 10-fold cross-validation scheme, as commonly validated for DBII [53]. Unfortunately, as shown in the central column, both approaches (i.e., *Ind-AR*) result in the worst bi-class classification scenario and harm average over all individuals rather than improving the performance, as expected. This effect is contrary to the case for subject-dependent preprocessing, which improves classifier performance for all FC metrics considered (see right column). A noteworthy fact is that the mean values of performance achieved by the power-based FC estimators outperform the corresponding phase-based metrics. As a result, GFC yields the best mean estimates of all feature extraction methods, while PLV comes in last.

### 4.2. Enhanced Performance of Individuals with Poor Skills

Next, we assess the influence of the proposed *SD-AR* preprocessing on the subjects executing the MI tasks the worst. As the GCF measure has the best performance (see Table 2), the analysis will be based on such an FC and conducted just for the DBII collection, testing many more individuals to obtain results with statistical significance. Moreover, we evaluate the methodology for enhancing Motor Imagery tasks’ classifier performance in subjects with poor coordination skills. In particular, we pay attention to inter and intra-subject variability, which may induce inferior performance of MI-based brain-computer interface systems.

By analyzing each performance measure (ACC, kappa, and AUC) individually, we cluster the separateness in terms of the variability between subjects. Concretely, clustering is carried out based on the classifier measures through the *k*-means algorithm, fixing the number of partitions to three, as commonly adjusted [54]. For evaluation purposes, the *k*-means algorithm is fitted with the performance criteria estimated for the feature set without processing (i.e., *Raw* scenario).

According to the considered training scenarios of MI classification, the classifier performances (mean and variance values) are grouped into the following three partitions of skills in practicing MI tasks (see Figure 5), as also recommended in [55]: (i) Group of individuals achieving the most consistent performance with very low variability of neural responses (colored in green). (ii) Group with adequate performance containing some response fluctuations (yellow color). (iii) Group producing modest performance and a high unevenness of responses (red color).

Figure 5a displays the resulting accuracy obtained by each training scenario. For the aim of assessing the gain in performance, the individuals are sorted based on the accuracy estimated by the LDA algorithm for the *Raw* training scenario. Moreover, Figure 5b shows the resulting partitions of MI skills. As can be observed, the baseline scenario results in the most reduced number of best-performing subjects (11). At the same time, this training with raw data yields the highest number of worst-performing individuals (namely, 21) that comprises more than 40% of the subject set. MI training under these conditions becomes very ineffective and costly to implement [56]. Afterward, the *Ind-AR* scenario delivers even fewer best-performing subjects (10) while reducing the worst-performing set by upgrading six individuals. Nonetheless, some subjects are severely downgraded, as is the case for #42 and #40.

Lastly, the proposed *SD-AR* approach results in the most extensive best-performing set (14) while providing the least number of individuals with poor skills (13). In other words, *SD-AR* reduces the group of poor-performing individuals from 40% to 25%. Another aspect to highlight is that *SD-AR* does not downgrade individuals compared with the other training scenarios.

Table 3 summarizes the classifier measures performed by each group, showing once more that *SD-AR* overperforms in all scenarios of preprocessing evaluated for artifact removal. *SD-AR* has also performed better than the average achieved by *Ind-AR* for all individuals.

### 4.3. Improved Interpretability of Elicited MI Responses

As part of the proposed *SD-AR* approach, we will consider the enhanced interpretability of GFC extraction to decode the brain neural responses elicited by MI paradigms. Here, we compute the topoplots based on the absolute value of individual LDA coefficients estimated within all four bandwidths and normalized between [0,1]. In this context, the absolute value of LDA coefficients is a measure of the relevance of discriminating between MI tasks, i.e., the larger the coefficient, the higher the contribution provided by the extracted FC set. Thus, to compute the relevance of each individual, we first build a supervector by concatenating all ravel versions of the LDA-based discriminant weights concerning the upper triangular GFC feature matrix. Concatenation integrates all four rhythms and produces a vector of size 4C×(C−1)/2. Nevertheless, since unsplit versions of μ and β are conventionally explained, the three subbands of the latter rhythm are merged into one plot using the maximum operator. Then, the newly recomputed relevance weight vector has a dimension of C×(C−1).

Afterward, we compute the group-level GFC representation by joining the relevance-weight vectors estimated for all $M_{i}$ individuals belonging to each *i*-th subset of MI skills (i.e., G I, G II, and G III), yielding a matrix with dimension $M_{i}\times C(C\;-\;1)$, being $M_{i}$ the number of subjects in *i*-th subset. The group-level relevance GFC matrix summarizes the joint contribution, from which the maximum value in each column (that is, per individual) becomes the connectivity link having the most significant contribution to discriminating between MI tasks.

Figure 6 displays the topoplots that reveal the estimated spatial relevance for μ and β rhythms. As can be seen, the fast changes of β enable a higher discriminating ability of neural responses than the μ band. Still, the assessed contribution of the elicited neural responses depends on the evaluated scenario of classifier training. Thus, the case of no removal filtering leads to brain β responses with background activity spreading the sensory-motor and occipital regions. However, the number of relevant GFC links decreases as the group performance of subjects also worsens. With both artifact subtraction filters applied across the entire group of individuals, significantly more β activity spreads across the scalp, overextending the sensory-motor cortex and including the brain activity coming from occipital and temporal lobes. Both regions are not supposed to contribute to the MI paradigm, and this activation may explain the worst classifier performance of *Ind-Ar* among the evaluated training scenarios. Regarding the proposed *SD-AR* approach, the discriminating neural responses are the most localized within the premotor and motor cortex. Therefore, the classifier is more accurate since more contributing GFC relationships are evaluated.

Furthermore, to decrease training efforts and encourage user-centered MI responses, the neurophysiological explanation of spatial patterns generated is enhanced by *SD-AR*. In particular, the estimated topoplots show that in comparison with the μ rhythm, β waveforms having a faster behavior benefit more from the artifact removal of the electrooculography and volume conduction effect. Consequently, the discriminating neural responses estimated by *SD-AR* are the most localized within the premotor and motor cortex, as expected in MI paradigms. To illustrate this finding, Figure 7 compares the corresponding plots of both rhythms estimated for the subject labeled as # sbj45 that is included by the *Raw* training scenario in the worst-performing group. After applying *SD-AR*, this subject achieves a high gain in classifier accuracy (more than 8%) so that he is newly incorporated into G II with adequate performance. As can be observed, the brain neural activity of β is now localized mainly in the sensorimotor area.

### 4.4. Method Comparison Results for MI Classification

The suggested preprocessing approach entails functional connectivity measures for feature extraction that encode most meaningful relationships of elicited brain neural responses between electrodes to tackle better MI tasks performed by poor-performing subjects. Several widely known power-based and phase-based methods of FC are tested, showing that subject-dependent preprocessing improves the classifier’s performance, with Gaussian Functional Connectivity being the most accurate. As a result, compared with several state-of-the-art methods recently reported for classification of MI tasks, Table 4 shows that the proposed *SD-AR* approach can achieve a competitive classifier accuracy for both EEG databases evaluated. In addition, the *SD-AR* approach decreases the group of poor-performing individuals, improving the use of MI paradigms. Still, the validation of DBII reveals that very few subjects may not benefit from the suggested subject-dependent preprocessing for artifact removal.

## 5. Concluding Remarks

We develop an approach for subject-dependent preprocessing to reduce the low SNR phenomenon in subjects performing poor motor imagery tasks by selectively removing artifacts combined with feature extraction using Functional Connectivity measures. As a result of the evaluation to remove electro-oculography artifacts and volume conduction effects, the proposed *SD-AR* approach improved the performance of the motor imagery classifier in subjects with poor motor skills. We investigate the reduction of two common artifacts in EEG data (electrooculography and volume conduction) that distort the extraction of feature sets and, thus, severely degrade the classifier performance of MI tasks. Surface Laplacian and Independent Component Analysis have been reported as two main preprocessing procedures for removing them. Still, their effectiveness strongly depends on the low-SNR phenomenon that becomes especially weak in subjects with poor skills in practicing MI tasks. To cope with this issue, we propose a selective subject-dependent preprocessing, termed *SD-AR*, that switches the application of either procedure depending on the classifier accuracy performed by each individual. The validation results obtained in the evaluated real-world databases indicate that the proposal overperforms in all scenarios of preprocessing evaluated for artifact removal. Nevertheless, even though neither preprocessing method requires any subject-specific parameter tuning, its effectiveness is enhanced as the EEG montage dimensions increase.

The authors plan to evaluate the developed approach for subject-dependent preprocessing in databases containing more subjects and lower-quality EEG data for future work. Furthermore, to reduce the low SNR phenomenon in subjects performing poor motor imagery tasks, more elaborate feature extraction methods are to be investigated. One more issue of consideration is to explore the effectiveness of the developed approach for artifact removal in modern architectures of deep learning [15]. Moreover, eye-tracking and cognitive psychological attention test data could be of benefit to enhance our artifact removal approach and favor both BCI performance and interpretability [66,67].

## Figures and Tables

**Figure 1 sensors-22-05771-f001:**
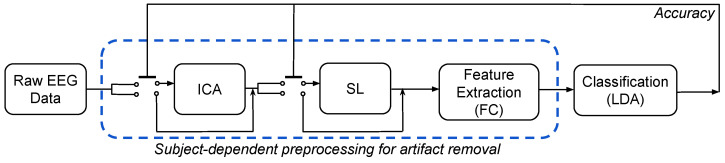
Diagram of the evaluated preprocessing approach to remove the effect of electrooculography and volume conduction. The dashed box highlights the proposed subject-dependent procedure for artifact removal (*SD-AR*) tested in subjects with poor MI skills.

**Figure 2 sensors-22-05771-f002:**
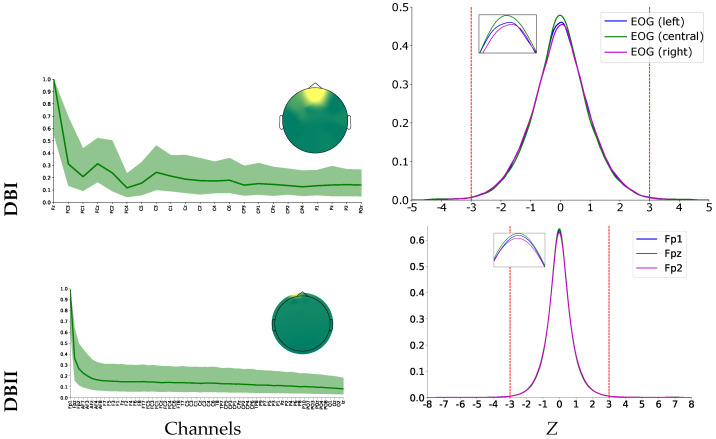
Effect of ICA filtering for EOG removal on both evaluated databases: DBI (**Top row**) and DBII (**bottom row**). Left column: the estimated electrode weights (normalized values are shown) and topograms. Right column: 3sigma-level threshold of similarity used for EOG removal.

**Figure 3 sensors-22-05771-f003:**
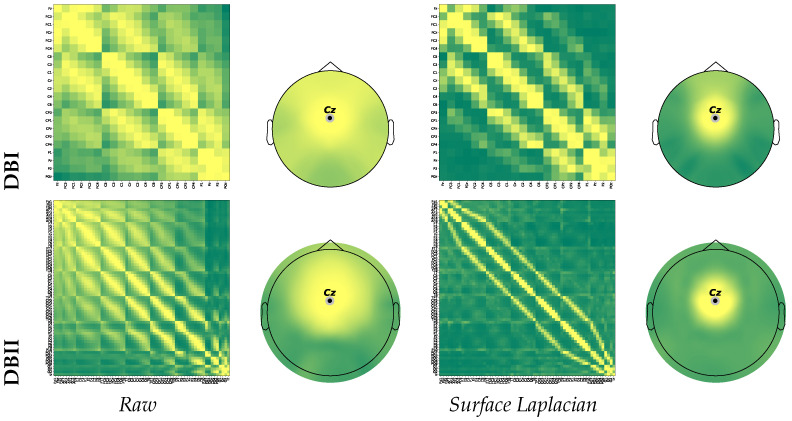
Effect of SL on the FC computation for DBI (**Top row**) and DBII (**bottom row**). Spatial FC matrix and the corresponding topoplot calculated without filtering (**Left column**); after SL filtering (**Right column**).

**Figure 4 sensors-22-05771-f004:**
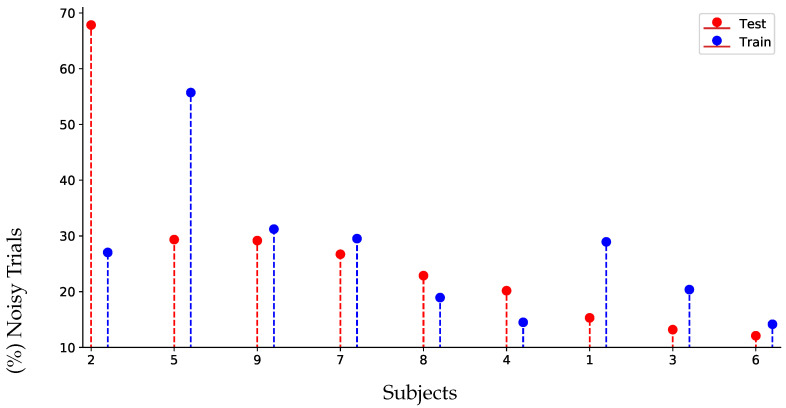
Percentage of noisy trials of each subject identified by the ICA-based artifact removal strategy in the training and testing stage of DBI.

**Figure 5 sensors-22-05771-f005:**
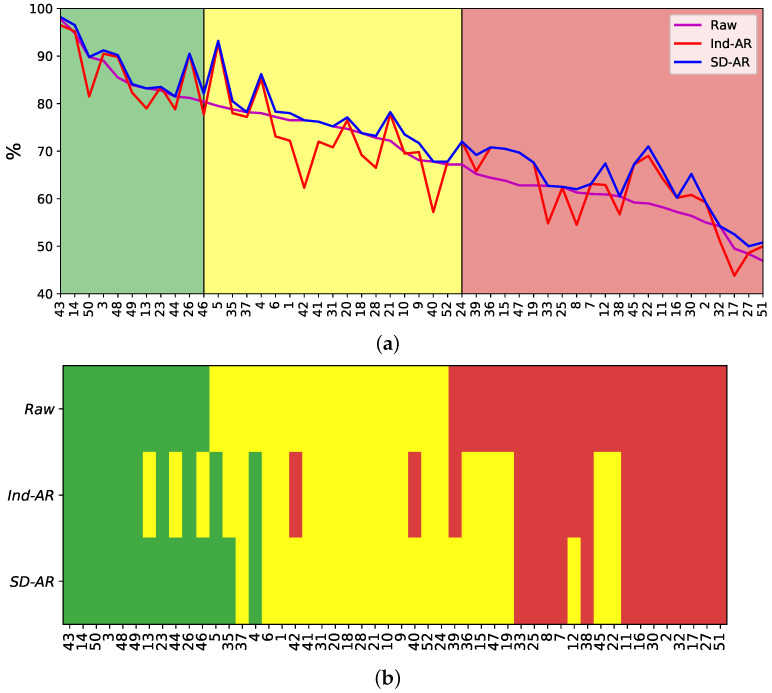
Clustering variability of individuals that belong to Group I (cells in green), Group II (yellow), and Group III (red), depending on the preprocessing strategy. (**a**) LDA-based Accuracy of MI tasks. (**b**) Resulting partitions of MI skills.

**Figure 6 sensors-22-05771-f006:**
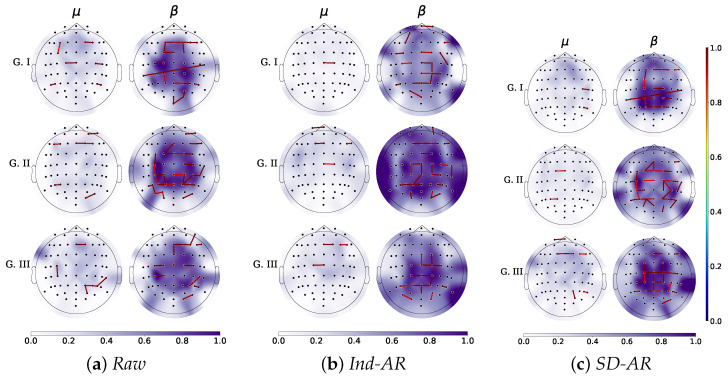
Topoplots show the GFC’s spatial relevance for each considered group of skill partition. The GFC links only exceeding relevance weight values of 0.9 are depicted. The strength values over each electrode estimate the background activity elicited by MI responses.

**Figure 7 sensors-22-05771-f007:**
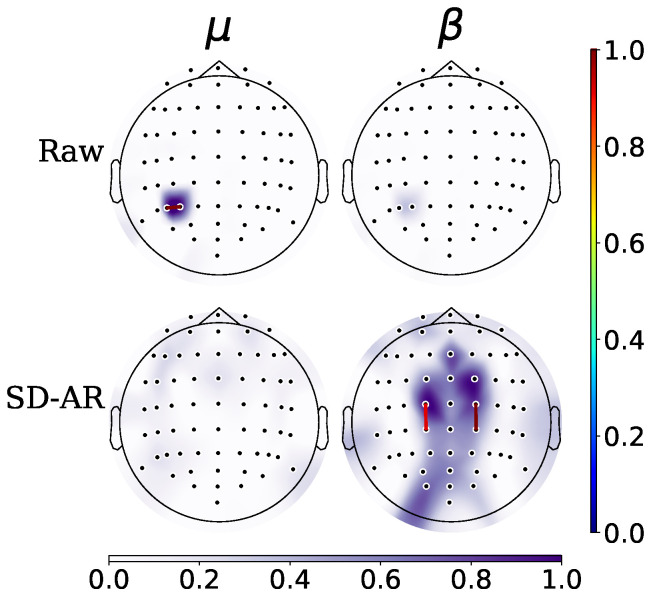
Topographical maps of # sbj45 with poor skills, having a high gain in classifier accuracy provided by *SD-AR*.

**Table 1 sensors-22-05771-t001:** The performance of each FC measure in DBI for each of the training classifier scenarios considered. The last row shows the performance values averaged over the whole subject set and across the evaluated FC measures. The notation PLV stands for Phase-Locking Value, COH—Spectral Coherence, Pearson—Pearson correlation, GFC Gaussian Functional Connectivity. FC: Functional connectivity. Bold stands for the best results.

FC	*Raw*	*Ind-AR*	*SD-AR*
	ACC Kappa AUC	ACC Kappa AUC	ACC Kappa AUC
PLV	44.9±8.8 0.27±0.12 0.72±0.09	53.3±11.9 0.38±0.16 0.77±0.10	51.1±12.9 0.35±0.17 0.76±0.11
COH	53.6±12.4 0.38±0.17 0.78±0.10	57.9±11.3 0.44±0.15 0.80±0.09	55.8±12.6 0.41±0.17 0.79±0.09
Pearson	54.6±12.4 0.39±0.17 0.79±0.10	57.4±11.8 0.43±0.16 0.81±0.09	57.2±11.7 0.43±0.16 0.81±0.09
Motifs	48.6±9.8 0.32±0.13 0.75±0.09	55.4±12.0 0.41±0.16 0.80±0.10	53.8±13.2 0.38±0.18 0.79±0.10
GFC	62.9±11.6 0.51±0.15 0.85±0.09	64.1±11.9 0.53±0.16 0.85±0.09	63.5±11.4 0.51±0.15 0.85±0.09
Average	52.9±12.6 0.37±0.17 0.78±0.10	57.7±12.4 0.44±0.16 0.81±0.09	56.3±13.1 0.42±0.17 0.80±0.19

**Table 2 sensors-22-05771-t002:** The performance of each FC measure in DBII for each of the training classifier scenarios considered. The last row shows the performance values averaged over the whole subject set and across the evaluated FC measures. FC: Functional connectivity. Bold stands for the best results.

FC	*Raw*	*Ind-AR*	*SD-AR*
	ACC Kappa AUC	ACC Kappa AUC	ACC Kappa AUC
PLV	62.1±9.7 0.24±0.19 0.62±0.10	61.8±12.8 0.23±0.26 0.61±0.13	66.7±11.4 0.33±0.23 0.67±0.11
COH	64.0±9.4 0.28±0.19 0.64±0.09	63.2±11.3 0.26±0.23 0.63±0.11	67.6±10.6 0.35±0.21 0.67±0.11
Pearson	65.0±10.5 0.30±0.21 0.65±0.11	63.9±12.0 0.27±0.24 0.63±0.12	68.4±11.1 0.36±0.22 0.68±0.11
Motifs	64.5±10.4 0.29±0.21 0.64±0.10	63.7±12.5 0.27±0.26 0.63±0.13	68.3±11.6 0.36±0.23 0.68±0.12
GFC	70.2±11.9 0.40±0.24 0.70±0.12	70.6±12.3 0.41±0.25 0.70±0.12	73.4±11.6 0.47±0.23 0.73±0.12
Average	65.2±10.8 0.30±0.22 0.65±0.11	64.6±12.6 0.29±0.25 0.64±0.13	68.9±11.5 0.37±0.23 0.69±0.12

**Table 3 sensors-22-05771-t003:** The resulting performance of each training classifier scenario obtained by GFC for each considered group of skills in practicing the MI tasks. FC: Functional connectivity. Bold stands for the best results.

FC	*Raw*	*Ind-AR*	*SD-AR*
	ACC Kappa AUC	ACC Kappa AUC	ACC Kappa AUC
G I	86.4±5.50 0.73±0.11 0.86±0.06	85.9±6.50 0.72±0.13 0.86±0.06	88.3±5.50 0.76±0.11 0.88±0.05
G II	73.9±4.10 0.47±0.08 0.74±0.04	72.8±7.80 0.45±0.16 0.72±0.08	76.5±5.90 0.52±0.12 0.76±0.06
G III	58.7±5.10 0.17±0.10 0.58±0.05	60.6±7.60 0.21±0.15 0.61±0.08	63.0±6.40 0.26±0.13 0.63±0.07

**Table 4 sensors-22-05771-t004:** Comparing the *SD-AR* classification accuracy of MI tasks with existing state-of-the-art methods reported for DBI (four classes) and DBII (bi-class). Bold stands for the best results.

	Method	Accuracy [%]
DBI	STR [57] MCSP+SRSG-FasArt [58] 3DCNN [59] EEGNet [60,61] SD-AR	49.2±15.6 62.7±15.9 64.9±7.5 67.3±15.7 63.5±11.4
DBII	FBCSP [62,63] EEGNet [60,63] RSTNN [63,64] Optical+ [65] SD-AR	68.0±15.0 64.0±7.0 69.0±12.0 69.6±16.3 73.4±11.6

## Data Availability

Datasets are publicly available at: http://www.bbci.de/competition/iv/index.html (accessed on 1 April 2022), and http://gigadb.org/dataset/100295 (accessed on 1 April 2022).

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
