# Peer review of "Subject-Dependent Artifact Removal for Enhancing Motor Imagery Classifier Performance under Poor Skills"

_sensors, 2022, doi:10.3390/s22155771_

Round 1

Reviewer 1 Report

I enjoyed reading the paper. The authors have presented an interesting article about a Subject-dependent Artifact Removal for Enhancing Motor Imagery Classifier Performance under Poor Skills.

Abstract, overview
The abstract is a concise description of the work. The introduction is well structured, and it covers all the concepts investigated in the methodological part. The previous work is well presented and integrated. I consider that this work brings added value in the field and the specific objectives of the manuscript are well related to the previous work developed in this domain. 

Methodology
The research design used is appropriate in order to answer the research questions proposed by the authors. The methods are described properly. The results are clearly presented and are in relation to the concepts investigated.

Discussion and conclusions
The discussions are clear and concise. The conclusions are strongly related to the findings of the research work.

Format and style
All the format and style features were respected and are compliant with the requirements.

References
The format of the reference list fixes well to the specified format.

Plagiarism and any other ethical concerns about this study
I do not have any potential conflict of interest with regards to this paper.

Despite the good work done, there is still some room for improvement, as follows:

  • I think some more literatures should be added. Besides the mentioned HMI there are several other systems (like cost-effect BCI, eye-tracking) which are applied nowadays. It would be good to see the "effect of different web-based media" content on "human brain waves", as well as the additional applications of brainwave-based control like in examine the effect of different web-based media on human brain waves. It would improve the quality of the publication to mention the relationship between a cognitive psychological attention test and the attention levels determined by a BCI systems such as in an examination and comparison of the EEG based attention test with CPT and TOVA. In addition to BCI systems, mentioning other important human-computer interaction eye movement tracking would also improve quality, as such systems can be used in the analysis of programming technologies such as LINQ and algorithms, thus enabling, for example, cognition load or source code, algorithm description tools readability testing like in measuring cognition load using eye-tracking parameters based on algorithm description tools, in clean and dirty code comprehension by eye-tracking based evaluation using GP3 eye tracker and in analyse the readability of LINQ Code using an eye-tracking-based evaluation.

Reviewer 2 Report

see attached file

Round 2

Reviewer 1 Report

I think some more literatures should be added. Besides the mentioned HMI there are several other systems (like cost-effect BCI, eye-tracking) which are applied nowadays. It would be good to see the "effect of different web-based media" content on "human brain waves", as well as the additional applications of brainwave-based control like in examine the effect of different web-based media on human brain waves. It would improve the quality of the publication to mention the relationship between a cognitive psychological attention test and the attention levels determined by a BCI systems such as in an examination and comparison of the EEG based attention test with CPT and TOVA. In addition to BCI systems, mentioning other important human-computer interaction eye movement tracking would also improve quality, as such systems can be used in the analysis of programming technologies such as LINQ and algorithms, thus enabling, for example, cognition load or source code, algorithm description tools readability testing like in measuring cognition load using eye-tracking parameters based on algorithm description tools, in clean and dirty code comprehension by eye-tracking based evaluation using GP3 eye tracker and in analyse the readability of LINQ Code using an eye-tracking-based evaluation.

Reviewer 2 Report

No further comments
